# Association of academic stress with sleeping difficulties in medical students of a Pakistani medical school: a cross sectional survey

Ahmed Waqas, Spogmai Khan, Waqar Sharif, Uzma Khalid and Asad Ali

CMH Lahore Medical College and Institute of Dentistry, Lahore Cantt, Pakistan

## ABSTRACT

**Introduction.** Medicine is one of the most stressful fields of education because of its highly demanding professional and academic requirements. Psychological stress, anxiety, depression and sleep disturbances are highly prevalent in medical students.
**Methods.** This cross-sectional study was undertaken at the Combined Military Hospital Lahore Medical College and the Institute of Dentistry in Lahore (CMH LMC), Pakistan. Students enrolled in all yearly courses for the Bachelor of Medicine and Bachelor of Surgery (MBBS) degree were included. The questionnaire consisted of four sections: (1) demographics (2) a table listing 34 potential stressors, (3) the 14-item Perceived Stress Scale (PSS-14), and (4) the Pittsburgh Quality of Sleep Index (PSQI). Logistic regression was run to identify associations between group of stressors, gender, year of study, student's background, stress and quality of sleep.
**Results.** Total response rate was 93.9% (263/280 respondents returned the questionnaire). The mean (SD) PSS-14 score was 30 (6.97). Logistic regression analysis showed that cases of high-level stress were associated with year of study and academic-related stressors only. Univariate analysis identified 157 cases with high stress levels (59.7%). The mean (SD) PSQI score was 8.1 (3.12). According to PSQI score, 203/263 respondents (77%) were poor sleepers. Logistic regression showed that mean PSS-14 score was a significant predictor of PSQI score (OR 1.99, $P < 0.05$).
**Conclusion.** We found a very high prevalence of academic stress and poor sleep quality among medical students. Many medical students reported using sedatives more than once a week. Academic stressors contributed significantly to stress and sleep disorders in medical students.

## INTRODUCTION

Medicine is one of the most stressful fields of education because of its highly demanding professional and academic requirements. Extensive medical curricula, frequent examinations and fear of failure are sources of constant stress and anxiety for medical students (*Shah et al., 2010*), who may cut short their leisure activities and hours of sleep in order to achieve their desired goals. Several studies have reported a high incidence of stress

Corresponding author
Ahmed Waqas,
ahmedwaqas1990@hotmail.com

disorders among medical students. According to a systematic review published in 2006, US and Canadian medical students suffer from a higher incidence of psychological distress, anxiety, depression and suicidal ideation than the general population (*Dyrbye, Thomas & Shanafelt, 2006*). *Sherina, Rampal & Kaneson (2004)* reported that 41% of Malaysian medical students suffered from psychological stress, which correlated directly with depressive symptoms. Several studies have reported the prevalence of psychological stress in medical students of different nationalities. Prevalence of stress was reported to be 20.9% in a Nepali medical school (*Sreeramareddy et al., 2007*), 63.8% in a Saudi Arabian (*Abdulghani et al., 2011*) and 90% in a Pakistani medical school (*Shaikh et al., 2004*). Psychological stress among medical students may have deleterious consequences and it further leads to poor academic performance (*Stewart et al., 1999*), sleep disorders (*Lemma et al., 2012*), alcohol and substance abuse (*Ball & Bax, 2002*), decreased empathy, a poor attitude towards the chronically ill, and cynicism (*Crandall, Volk & Loemker, 1993*).

Several studies have also reported poor sleep quality in medical students. *Anjum, Bajwa & Saeed (2014)* found that the prevalence of disturbed sleep patterns among Pakistani medical students was higher than in their non-medical counterparts. Psychological stress is a triggering factor for insomnia and has a bidirectional association with poor sleep quality (*Suchecki, Machado & Tiba, 2009*). Thus, it represents a vicious cycle that is associated with adverse mental health consequences in medical students. Poor sleep quality is also associated with dysregulation of the human metabolome (*Davies et al., 2014*), high failure rates and poor academic performance (*Curcio, Ferrara & De Gennaro, 2006*).

Most of the medical schools in Punjab, Pakistan offer a 5-year-long Bachelor of Medicine and Bachelor of Surgery (MBBS) degree program divided into 2 preclinical years followed by 3 clinical years. These schools use conventional non-problem-based learning (PBL) teaching methods, which consist of teacher-centered didactic techniques, long lectures, tutorials and practical tasks. Students are frequently assessed with written, oral or practical examinations throughout the year, and each year ends with a final exam held by the University of Health Sciences in Lahore. A passing grade on this exam is necessary to qualify for the next academic year. According to *Shah et al. (2010)*, Pakistani medical students who reported academic stressors as often/always were 3.45 times more likely to be suffering from psychological stress than their counterparts reporting a never/sometimes response for it.

Several studies have reported the prevalence of stress among medical students in Pakistani medical schools. However, the paucity of knowledge on pattern of sleeping difficulties in Pakistani medical students and its association with academic stress warranted this study. This study was designed with two aims: (1) to determine the prevalence of psychological stress and pattern of sleeping difficulties among medical students, and (2) to explore the relationship between academic stressors, psychological stress and poor sleeping habits among medical students.

## METHODOLOGY

### Study design

This cross-sectional study was done at Combined Military Hospital Lahore Medical College (CMH LMC) and Institute of Dentistry, Lahore (Pakistan) after approval by the CMH LMC Research Ethics Committee. Students from all yearly courses of the MBBS degree program were included in this study, which took place from August, 2014 to September, 2014. The sample size required for this survey was calculated as 251 participants for a 95% confidence level, a 5% margin of error and a population size of 720 medical students. Students were randomly selected with a stratified sampling technique. First, the students were divided into groups based on their year of study, then random numbers were generated using computer software. A total of 280 self-administered questionnaires were distributed to potential respondents to ensure an adequate percentage response rate. Written informed consent was obtained from all participants. They were informed about the aims of this study and assured anonymity and that only group-level findings would be reported.

### Questionnaire

The questionnaire consisted of four sections: (1) demographics (2) a table listing 34 potential stressors, (3) the 14-item Perceived Stress Scale (PSS-14), and (4) the Pittsburgh Quality of Sleep Index (PSQI). The demographics section recorded data for participants' age, gender, year of study, residence and background. The table of potential stressors was taken from a similar study in a Pakistani medical school by *Shah et al. (2010)* and used to obtain data on the frequency and severity of each potential stressor. Frequency was reported as never, rarely, sometimes, often or always, and severity was rated with a Likert-type response scale of increasing severity from 1 to 10.

The PSS-14, developed by *Cohen, Kamarck & Mermelstein (1983)*, is one of the most widely used psychological instruments to measure perceived stress. It comprises 7 positively-stated and 7 negatively-stated items, and is scored by reverse coding the negatively-stated items and then summing the scores for all 14 items. The scores range from 0 to 56 with higher scores indicating higher levels of stress.

The fourth section of the questionnaire consisted of the PSQI, a self-rating questionnaire that assesses sleep quality and disturbances during the preceding month. This instrument comprises 19 individual items and yields an overall score ranging from 0 to 21. Participants who score higher than 5 are considered poor sleepers. It also yields scores on seven other components: subjective sleep quality, sleep latency, sleep duration, habitual sleep efficiency, sleep disturbance, use of sleeping medication and daytime dysfunction (*Buysse et al., 1989*).

### Data analysis

Al data were analyzed in SPSS Inc., (Chicago, Illinois, USA) v.20 software. Frequencies were calculated for demographic variables. Mean PSS-14 scores were calculated and divided into quartiles, then further recoded as a dichotomous variable by combining the upper

two quartiles and lower two quartiles (cut-off value = 28). These values were recoded as 1 = high stress levels and 0 = low stress levels, as in an earlier study from Pakistan (*Shah et al., 2010*). The frequencies of stressors were grouped into dichotomies as follows: never/rarely/sometimes = 0 and often/always = 1. These stressors were also grouped into three domains: psychosocial stressors, academic stressors and health-related stressors. Mean severity of stressors was calculated. Logistic regression was used to identify the determinants of perceived stress. The PSS-14 score was used as a dependent variable and age, year of study, gender, residence, background, psychosocial stressors, health-related stressors and academic stressors were used as independent variables. Response frequencies and percentage were reported for subscales of PSQI. Binary logistic regression was used to identify associations between PSS-14 scores and PSQI global scores (dichotomous variable). Independent sample *T* test was run to analyze association between academic stressors (less than often/often and always) and PSQI scores.

## RESULTS

### Demographic characteristics

Total response rate was 93.9% (263 of 280 respondents returned the questionnaire). Mean age (SD) of respondents was 21.1 years (1.78). The gender distribution of the respondents was 148 women (56.3%) vs. 115 men (43.7%). Most participants were of Pakistani background (220, 83.7%) and the remaining 43 respondents (16.3%) were of other nationalities. Most of the participants resided off the medical school campus (161, 61.2%) and the remaining 102 (38.2%) resided in student residence facilities. The distribution according to year of study was 1st year 58 (22%), 2nd year 67 (25.5%), 3rd year 48 (18.3%), 4th year 45 (17.1%) and 5th year 45 participants (17.1%).

### Perceived stress

Mean (SD) PSS-14 score was 30 (6.97). According to logistic regression analysis, cases of high-level stress were associated with year of study and academic stressors only (Table 1). Univariate analysis identified a total of 157 cases with high stress levels (59.7%). Most respondents with high levels of stress were 2nd-year students (48, 71.6%) and 5th-year students (32, 71.1%), followed by 1st-year (29, 50%), 3rd-year (24, 50%) and 4th-year students (24, 53.3%).

### Stressors

The frequency and severity are shown for academic stressors in Table 2, psychosocial stressors in Table 3, and health-related stressors in Table 4. The frequency of different stressors was reported as often/always by 147 (55.9 %) respondents for academic stressors, 45 (17.1%) for psychosocial stressors, and 48 (18.3%) for health-related stressors.

### Quality of sleep

Mean (SD) PSQI score was 8.1 (3.12). According to these scores, 203/263 respondents (77.02%) were poor sleepers. Logistic regression showed that mean PSS-14 score was a significant predictor of PSQI score (OR 1.99, $P < 0.05$). During the month preceding

**Table 1 Determinants of stress according to logistic regression analysis.** $R^2 = 0.095$ (Cox & Snell), 0.128 (Nagelkerke). Model chi-squared value = 26.258 (df = 11).

| Determinants | Odds ratio (OR) | 95% CI for OR |
|---|---|---|
| Age | 0.880 | 0.651–1.190 |
| Gender | | |
| Female | 1 | |
| Male | 1.339 | 0.748–2.397 |
| Residence | | |
| National | 1 | |
| International | 0.914 | 0.508–1.645 |
| Residence | | |
| Off-campus residence | 1 | |
| On-campus residence | 1.291 | 0.606–2.750 |
| Year[*] | | |
| 1st | 0.197 | 0.039–.999 |
| 2nd | 0.563 | 0.151–2.11 |
| 3rd | 0.291 | 0.086–0.987 |
| 4th | 0.347 | 0.126–0.957 |
| 5th | 1 | |
| Psychosocial stressors | 1.231 | 0.571–2.652 |
| Academic stressors[**] | 2.470 | 1.424–4.284 |
| Health-related stressors | 0.818 | 0.389–1.721 |

**Notes.**

[*] $P < 0.05$.

[**] $P < 0.01$.

**Table 2 Frequency and severity of academic stressors.**

| Academic stressors | Often/Always response, $n$ (%) | Severity (1—lowest to 10—highest) |
|---|---|---|
| Exam frequency | 190 (73.6) | 7 |
| Academic performance | 148 (57.4) | 7 |
| Academic curriculum | 125 (48.4) | 6 |
| Dissatisfaction with class lectures | 86 (33.3) | 6 |
| Unavailability of learning materials | 55 (21.3) | 5 |
| Becoming a doctor | 106 (41.1) | 6 |
| Lack of leisure time | 123 (47.7) | 6 |
| Competition with peers | 77 (29.8) | 5 |
| Performance in practical | 73 (28.3) | 5 |
| Lack of special guidance from faculty | 70 (27.1) | 5 |

**Peer**J

**Table 3** Frequency and severity of psychosocial stressors.

| Psychosocial stressors | Often/Always response, $n$ (%) | Severity of stressors (1—lowest to 10—highest) |
|---|---|---|
| High parental expectations | 137 (53.1) | 7 |
| Loneliness | 64 (24.8) | 5 |
| Family problems | 62 (24) | 5 |
| Living away from home | 66 (25.6) | 6 |
| Political situation of country | 42 (16.3) | 5 |
| Relations with opposite sex | 41 (15.9) | 5 |
| Difficulty reading textbooks | 51 (19.8) | 5 |
| Lack of entertainment in Lahore | 71 (27.5) | 5 |
| Difficulty with the journey back home | 45 (17.4) | 5 |
| Quality of cafeteria food | 100 (38.8) | 7 |
| Financial strain | 49 (19) | 5 |
| Inability to socialize with peers | 26 (10.1) | 7 |
| Living conditions in student residence | 49 (19) | 6 |
| Member of fraternity or sorority | 14 (5.4) | 5 |
| Lack of personal interest in medicine | 28 (10.9) | 5 |
| Adjustment with roommate | 37 (14.3) | 5 |

**Table 4** Frequency and severity of health-related stressors.

| Health-related stressors | Often/Always response, $n$ (%) | Severity (1—lowest to 10—highest) |
|---|---|---|
| Power failures | 139 (53.9) | 7 |
| Difficulty sleeping | 101 (39.1) | 6 |
| Class attendance | 112 (43.4) | 7 |
| Nutrition | 75 (29.1) | 6 |
| Exercise | 77 (29.8) | 6 |
| Quality of cafeteria food | 85 (32.9) | 7 |
| Physical disability | 10 (3.9) | 5 |
| Substance abuse | 16 (6.2) | 5 |

the survey, 73 (27.8%) respondents got less than 5 h of sleep per day, 233 (88.6%) reported nighttime disturbances, 97 (36.9%) poor sleep latency, 128 (48.7%) daytime dysfunctioning, 94 (35.7%) poor sleep quality and 31 (11.8%) poor sleep efficiency. A few respondents 13 (4.9%) reported using sedatives more than once a week.

The chi-squared test revealed a significant association between stress and poor quality of sleep (chi-squared = 5.48, $P < 0.05$). The prevalence of poor quality of sleep among stressed students was 82% (129/157), whereas among non-stressed students, only 69.8% were poor sleepers. Independent sample $T$ test revealed a significant association in mean scores of PSQI scale and academic stressors. Those students who reported academic

stressors as often/always had a higher mean score on PSQI scale than their counterparts who reported academic stressors as never/sometimes (Mean difference = 1, $P < .05$).

## DISCUSSION

Our sample consisted of medical students enrolled at a privately financed Pakistani medical school which favors a strictly teacher-centered, non-PBL conventional teaching environment. Our results document a high prevalence of psychological stress (59.7%) and poor sleep quality (77%), which is in consonance with earlier studies conducted in Pakistan. Most of the respondents (55.9%) indicated that they often or always experienced academic stressors, and this subgroup was 2.5 times as likely to suffer from psychological stress and poor sleep quality as their non-stressed counterparts. These results underscore the importance of nurturing a positive learning environment for medical students.

An interesting finding in our analysis was that both male and female students were equally likely to experience stress and sleep disturbances. This result is consistent with a report by Cohen and colleagues, who found no significant association between stress levels and gender in a sample of college students (*Cohen, Kamarck & Mermelstein, 1983*). In 2004, a systematic review of anxiety and depression in the general community reported a higher prevalence of depression in women (mean point prevalence 45.5%) than in men (*Mirza & Jenkins, 2004*). Similarly, according to Shah and colleagues, female medical students were more likely to experience stress than male medical students (*Shah et al., 2010*). Our study was based at a privately financed medical school, where most of the medical students belong to higher socioeconomic classes and might be assumed to come from a more tolerant cultural background. Therefore, these students might not be exposed to the same psychosocial stressors as students from a more gender-sensitive and patriarchal environment. Medical students enrolled in their 2nd and 5th (final) year of the MBBS program experienced higher levels of stress. This might be due to the extensive teaching curriculum in the 2nd year, and due to the introduction of clinical subjects and clerkships in the final year of the MBBS degree program.

Another interesting finding in our study was that 77% of the medical students reported poor sleep quality, which was significantly associated with academic stressors. A high percentage of respondents (27.8%) got less than 5 h of sleep per night. This is in consonance with a large-scale study of 2,515 Ethiopian university students, 55.8% of whom reported poor sleep quality. That study also reported a strong association between poor sleep quality, stress, anxiety and depression (*Lemma et al., 2012*). Drug misuse to induce sleep was reported by 4.9% of the students. *Zafar et al. (2008)* in their survey of four universities in Karachi, Pakistan, found a high prevalence of self-medication by students. More than 33 (7.6%) of their participants reported insomnia as a reason for self-medication, and 44 (10%) participants admitted to the misuse of sleeping pills. The relatively high use of sleep-inducing medication may have been related with the fact that Pakistani pharmacies sell these drugs without a prescription. According to a 2005 survey, an alarming proportion of Pakistani pharmacies (50/311, 16.1%) reported selling this type of medication without a prescription (*Butt et al., 2005*). Our analysis showed that

94 (35.7%) respondents complained of daytime sleepiness. People with daytime sleepiness because of insomnia have lower self-esteem and are three times as likely to be involved in road accidents as their well-rested counterparts (*Garbarino et al., 2002*).

Our results are consistent with previous studies at medical schools in Nepal (*Sreeramareddy et al., 2007*) and Pakistan (*Shah et al., 2010*) that reported a strong association between academic stressors and psychological morbidity. Both of the schools involved in those studies had a conventional teaching environment. However, the stress associated with a conventional teaching environment can be reduced. The ultimate aim of medical education should be to produce competitive but compassionate, reflective, self-reliant and empathetic doctors. But the stressful environment of medical schools leads to "hardening of [the] heart during medical school" i.e., a decline in the capacity of medical students to empathize (*Newton et al., 2008*). Due to academic stress together with high academic and professional expectations, medical students also report suicidal ideation during their school years (*Hershner & Chervin, 2014*). But these statistics might be underreported in Pakistan because of public stigma revolving around psychiatric illnesses in medical students (*Waqas et al., 2014*). To address this grave situation, effective screening for psychological stress, anxiety and depression in medical students, along with psychotherapeutic and educational interventions, should be introduced at Pakistani medical schools. Long-term plans should be devised to introduce changes in the medical curriculum to make it less rigid and burdensome on students. New teaching methods should be implemented at Pakistani medical schools to make the learning environment more student-friendly and enjoyable. There is overwhelming evidence that students in the Indian subcontinent perceive positively and welcome PBL methods. A study that compared perceptions towards PBL experiences reported that medical students found these approaches to be better at enhancing team work, interpersonal relationships, motivation and personal enjoyment, and at favoring positive attitudes towards information-gathering, reasoning and independent thinking (*Nandi et al., 2000*). Students in a PBL curriculum reported better quality of life, perhaps because of the spare time available for self-study and the greater freedom and autonomy to manage their time (*Tempski et al., 2012*). A randomized controlled trial concluded that mindfulness-based stress reduction programs resulted in mental well-being and improved quality of life in medical and psychology students (*De Vibe et al., 2013*). In their meta-analysis, *Shapiro, Shapiro & Schwartz (2000)* provided overwhelming evidence of improvement in the psychological well-being of medical students who underwent stress reduction programs such as mindfulness-based stress reduction, hypnosis, desensitization, progressive muscle relaxation, social support and group therapy. Such programs also improve participants' spirituality, empathy, positive coping skills and conflict resolution skills (*Shapiro, Shapiro & Schwartz, 2000*). Studies should be done to determine the feasibility of campus-based psychological support, stress reduction and relaxation-based interventions such as mindfulness training at Pakistani medical schools.

The cross sectional design of this study limits inferences about causality and temporality between academic stressors, psychological stress and sleeping difficulties. Our sample size

consisted of a representative sample of a single medical school. Therefore, these results are not generalizable to whole student population in Pakistani medical colleges. The use of self-administered questionnaires is an important limitation in this study and it may lead to recall bias. Psychological stress and sleeping disturbances were assessed with psychometric instruments which are not completely transposable to the DSM diagnostic criteria for psychiatric illnesses.

## CONCLUSION

Our study revealed a high prevalence of academic stress and poor sleep quality among medical students in Lahore. Many medical students reported using sedatives more than once a week. Academic stressors contributed significantly to perceived stress and sleep disorders.

## ACKNOWLEDGEMENTS

The authors thank Aimen Haider, Shahbakht Ilyas, Sayyedah Khadeeja Bokhari and Asad Abbas, students at CMH Lahore Medical College, for their help with data collection. We also thank K Shashok (AuthorAID in Eastern Mediterranean) for improving the use of English in the manuscript.

### Funding

The authors declare there was no funding for this work.

### Competing Interests

The authors declare there are no competing interests.

### Author Contributions

- Ahmed Waqas conceived and designed the experiments, performed the experiments, analyzed the data, contributed reagents/materials/analysis tools, wrote the paper, prepared figures and/or tables, reviewed drafts of the paper.
- Spogmai Khan and Uzma Khalid conceived and designed the experiments, performed the experiments, wrote the paper, prepared figures and/or tables, reviewed drafts of the paper.
- Waqar Sharif performed the experiments, analyzed the data, wrote the paper, prepared figures and/or tables, reviewed drafts of the paper.
- Asad Ali conceived and designed the experiments, performed the experiments, analyzed the data, wrote the paper, prepared figures and/or tables, reviewed drafts of the paper.

### Human Ethics

The following information was supplied relating to ethical approvals (i.e., approving body and any reference numbers):

This study was approved by Ethical Review Committee of CMH Lahore Medical College, Lahore Cantt. Pakistan. An approval letter from CMH LMC ethical review committee has been provided.

## Supplemental Information

Supplemental information for this article can be found online at http://dx.doi.org/10.7717/peerj.840#supplemental-information.

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
