# Peer review of "Association of academic stress with sleeping difficulties in medical students of a Pakistani medical school: a cross sectional survey"

_PeerJ, doi:10.7717/peerj.840_

## Round 0.1 · original submission · Major Revisions

· Academic Editor

Major Revisions

Please, read all the reviewers’ comments and revise the manuscript accordingly.

Following the comments by Reviewer #1:

1. Check the accuracy of the title
2. Make sure that your introduction section clearly indicate the knowledge gap your study intends to fill.
3. Check the consistency of Tables 2 and 3.
4. Clarify the assumptions used for sample size calculation and check the accuracy of calculation.
5. Rephrase the statements on “controlling effects” to make it more clear what you mean by that.
6. Clarify how different models were established and if the model assumption have been checked.
7. You can disregard the comment that “the sample size and sampling procedure must be presented separate from the study design.”
8. Revise the discussion section according to the reviewer’s comments, especially avoid discussing the findings that were not reported in the Results section. Also, make sure to discuss the limitations of your study.

Following the comments by Reviewer #2:

1. Modify the manuscript according to all of the reviewer's comments under „Basic reporting“
2. Clarify which are the dependent and which are independent variables in lines 110-112
3. Address the reviewer's comment on the necessity of adaptation and validation of the questionnaire.
4. Erase the sentence in lines 212-215 and the accompanying reference.

Reviewer 1 ·

Basic reporting

The author tried to assess the level of stress and status of sleep quality among university students and their relationship with each other. However looking at the title of the manuscript, it is not clear what the author intended to capture. It is obvious that academic stressers could lead to stress that is why they are called stressers. Instead it would have been meaningful to talk about stress and sleep disturbance. The mediation effect is not clear and it is not captured in this document. In addition, the English has to be reviewed by expert as it has some problem.
the introduction section did not clearly show the gap this study tried to address. it did not have any rationale at the end.
the tables 2 and 3 needs revision as their contents are not consistent with their category names.

Experimental design

Methods section is not well organized and did not provide sufficient information to be reproducible. the sample size and sampling procedure must be presented separate from the study design.
The data analysis also requires some clarity; for instance, the analysis for mediation was not clearly described. It is not clear how the different models were established and if the model assumptions were checked.

Validity of the findings

Line 70-73: the author provides the assumptions used for sample size calculation. However it is not clear what the first 50% assumption was about. Given the assumption the final sample size should be 384 instead of 280. This has to be clearly justified. The sample size is not adequate even to measure the magnitude of the outcome variables let alone to establish a statistical significance between variables.
• In line 153: the author has used linear regression for analyzing controlling effect. It is not clear what controlling effect mean and I don’t think it is appropriate terminology even. The author must clearly describe why linear model was used in the method section.
The discussion did not focus on the key findings of the study. For instance, the first paragraph of the discussion section should summarize the key findings instead the point discussed here was already described somewhere in the document and not that relevant.
In the discussion section, only those results which were presented as findings in this study were supposed to be discussed however, there were findings which were not reported in the result section but discussed here. For instance, line 182-185; 191-196; 198-199 were all results not presented in the document anywhere but here.
In the middle of the discussion the author tried to make some kind of recommendation which is not acceptable; this will miss the purpose of discussion in scientific writing.
The author did not discuss any limitations of this study at all and this could be very misleading for someone using the study as a reference. this will make its validity questionable.
In the conclusion section, the author recommended possible curriculum change. None of the finding lead to such change given the cross-sectional nature of the study with inadequate sample size, the author should be cautious to make such recommendation.
the author might re-write the rest of the document but the major methodological issue which is the sample size, can not be solved at this time.

Additional comments

None

Annotated reviews are not available for download in order to protect the identity of reviewers who chose to remain anonymous.

·

Basic reporting

1. Re frame the sentences starting from line 29-37. These sentences lack clarity and does not fit seamlessly with each other
2. Sentence in line 32-34 points that other studies done in this area did not use DSM diagnostic criteria for psychiatric illnesses. The current study also does not use DSM criteria, nor it explores any psychiatric morbidity. In this background, that particular sentence does not fit in the introduction part
2.Sentence in line 51-53. Is it implied that Pakistani medical students have more stress than their non-medical counterparts, or than their counterparts from other parts of the world?
3. Paragraph from line 54-60. It is not appropriate here. Perhaps a mention of the gist of that paragraph could be added in discussion part.

Experimental design

It is mentioned that the questionnaire for potential academic stressors is an adaptation from the 2007 study of Sreeramareddy, 2007. However, it appears to be the very same questionnaire used in the 2007 study. There is a vast socio cultural and geographic difference between the Nepal and Pakistan, and hence proper adaptation and if possible cultural validation of the questionnaire of 2007 study has to be done before it could be used. It is also advised that the steps in adaptation and validation be described.

Validity of the findings

Sentence in 212-215 - the sentence is totally uncalled for. It is a scientific paper that is being written and let there be not any sensationalism in it. Moreover, it is unethical to to name the names.

110-112. The sentence is not clear about which are the dependent and independent variables.

Additional comments

The comments in all the 3 areas should be considered and appropriate changes should be made

---

## Round 0.2 · Minor Revisions

· Academic Editor

Minor Revisions

I believe your manuscript is worthy of publication in PeerJ after some minor revisions. I will send you Word documents with suggestions for further revisions separately (from my e-mail address). Although the Reviewer #2 still thinks that your study has a flawed methodology, I don't agree with her, so you don't have to address her comments.

·

Basic reporting

None

Experimental design

The questionnaire used for academic stressors, continues to be that of Sreeramareddy (2007) with no culture specific changes and adaptation done

There is no sense in trying to find association between perceived stress and stressors since essentially both are the same

Validity of the findings

Reflects the flaws in methodology

Additional comments

Methodology needs to be worked out . The tools should be culturally adapted and the authors should have clarity on the operational definitions of the variables under study, as well as regarding the tools

---

## Round 0.3 · Minor Revisions

· Academic Editor

Minor Revisions

One of my comments for the previous version was: "Please, round all the values (means, percentages) on one decimal place. Standard deviations, however, should be rounded on two decimal places. Apply this system throughout the manuscript." In the latest version, means are rounded on one decimal places and SDs on two, but some percentages are still rounded on two or even three decimal places (instead of one). Please, go again through the whole manuscript (including abstract) and be careful to round all the percentages to one decimal place.

---

## Round 0.4 · accepted · Accept

· Academic Editor

Accept

No further comments from my side.